# Identification and Molecular Characterization of Giant Liver Fluke (*Fascioloides magna*) Infection in European Fallow Deer (*Dama dama*) in Romania—First Report

**DOI:** 10.3390/microorganisms12030527

**Published:** 2024-03-06

**Authors:** Dan-Cornel Popovici, Gheorghe Dărăbuș, Ana-Maria Marin, Ovidiu Ionescu, Maria Monica Florina Moraru, Mirela Imre, Emil Tîrziu, Narcisa Mederle

**Affiliations:** 1Forestry Faculty, Transilvania University Brasov, Sirul Beethoven, 500123 Brasov, Romania; danpopovici30@yahoo.com (D.-C.P.); o.ionescu@unitbv.ro (O.I.); 2Faculty of Veterinary Medicine, University of Life Sciences “King Michael I” from Timisoara, 300645 Timisoara, Romania; gheorghe.darabus@fmvt.ro (G.D.); mariamoraru@usvt.ro (M.M.F.M.); mirela.imre@usvt.ro (M.I.); emiltirziu@usvt.ro (E.T.); narcisamederle@usvt.ro (N.M.)

**Keywords:** fascioloidosis, parasitic diseases, wild ruminants

## Abstract

Fascioloidosis is a parasitic disease of primary wild and domestic ruminants, caused by giant liver fluke, *Fascioloides magna*. The definitive host of the liver fluke in its area of origin (North America) is the white-tailed deer (*Odocoileus virginianus*). In Europe, the red deer (*Cervus elaphus*) and European fallow deer (*Dama dama*) are definitive hosts and the most sensitive hosts to *F. magna* infection, on which the parasite exerts serious pathogenic effects. In this study, we analyzed fecal samples and livers of 72 *D. dama* from 11 hunting grounds in Arad County, Romania. Of the 72 fecal samples and livers from *D. dama*, trematodes of the genus *Fascioloides* were identified in four (5.56%). Sequencing revealed that the trematodes identified in the samples were similar to the sequence of *F. magna* (GenBank no. EF534992.1, DQ683545.1, KU232369.1). The sequence obtained from the molecular analysis has been deposited in GenBank^®^ under accession number OQ689976.1. This study describes the first report of giant liver fluke (*F. magna*) infection in *D. dama* in Romania.

## 1. Introduction

Parasitism is a particularly complex phenomenon that involves the host and the parasite and the interactions established between them [1]. The nature of these interactions is dynamic, and it can be easily anticipated that both the host and parasite will try to adapt to this situation without being able to control the duration and end of this process [2]. The appearance of new species of parasites or spatial variations in parasite–host interactions (especially new, unusual hosts) are major factors affecting the monitoring of this co-evolutionary process [3].

One of the most important parasites of cervids and domestic ruminants, the giant liver fluke, *F. magna*, originally from North America, was introduced to Europe, in Italy, and was described by Bassi in 1875. The parasite conquered the region of Bohemia, the Czech Republic [4], and was identified in the forests along the banks of the river Danube. Nowadays, the giant liver fluke is present in Slovakia [5,6], Germany [7], Austria [8,9], Poland [10], Hungary [11], Croatia [12], and Serbia [13]. The adults of *F. magna* are large, thick, and fleshy, measuring up to approximately 3–10 × 2–3 cm. The oral and ventral suckers and some of the internal structures, particularly elements of the alimentary and reproductive systems, can easily be seen microscopically in fixed, stained specimens. Eggs of this trematode are oval, measure approximately 140 µ by 85 µ, and have a thin, smooth shell, an operculum (lid) at one end, and often a small “pimple” at the opposite end [1].

The definitive host of the liver fluke in North America is *O. virginianus* [14,15], and in Europe, there is a classification into definitive hosts, aberrant hosts, and dead-end hosts. In Europe, the first and most important group is represented by *C. elaphus* and *D. dama*, the most sensitive hosts to *F. magna* infection, on which the parasite exerts serious pathogenic effects [6,14,16]. *Fascioloides magna* infection, among other pathological factors, has a negative influence on the health, growth, and productivity of ungulates: hepatic pseudocysts and significant weight loss, affecting trophy and reproductive performance [16,17,18].

Although the aberrant hosts—mouflon (*Ovis ammon musimon*) [19], sheep (*Ovis aries*) [20], goats (*Capra aegagrus hircus*), rabbits (*Oryctolagus cuniculus*), guinea pigs (*Cavia porcellus*), and recently, roe deer (*Capreolus capreolus*) [18]—are not involved in trematode reproduction, their infection has fatal consequences [14,21]. Large ungulates—moose (*Alces alces*), American bison (*Bison bison*), sika deer (*Cervus nippon*), llama (*Lama glama*), wild boar (*Sus scrofa*) [22], horse (*Equus ferus caballus*), pig (*Sus domesticus*), and cattle (*Bos taurus*)—are considered as dead-end hosts [16,21,23,24,25,26]. The first report of an infection with *F. magna* in primates (*Cercopithecus petaurista*) was made in North America by Hasse K.E. et al., 2021 [27].

In correlation with the three outbreaks recognized in Europe (the La Mandria natural park in Italy, the south and center of Bohemia and the territory of the flooded forest of the Danube), the presence of only two species of snails, intermediate hosts *Galba truncatula* and *Radix* (syn. *Lymnaea*) *peregra*, has been confirmed [16,28]. *D. dama*, an important selenodont species since ancient times, disappeared from the country’s fauna during the glacial period, being later (re)introduced as an alien species by the Romanians through anthropochore dissipation [29]. In Romania, the *D. dama* is present in 93 hunting funds from 25 counties, reaching a herd of 7508 specimens evaluated in 2022 [30]. This population is surplus compared with the optimal level established for this species (3863 specimens) because of the assessment of the support capacity of the various types of habitats used by this animal [30].

*D. dama* prefers forests, alternating with open spaces, pastures, and agricultural lands located at an altitude varying from a few meters to 520 m (e.g., Reci-Lisnău hunting ground, Covasna County, Romania) [31]. The diversity of microclimates in which *D. dama* lives and breeds in Romania affirms its adaptability, which is certainly to the advantage of the species, considering both the dynamic action of climatic and anthropogenic changes and the variability of interspecific relationships, the host–parasite relationship being the most appropriate example [32]. Dicrocelian trematodes and paramphistomes [33], as well as gastrointestinal nematodes [34,35], have been identified in *D. dama* in Romania, but there have been no reports on parasitism with *F. magna* until now. The aim of this study was to investigate the occurrence of *F. magna* in *D. dama* and to molecularly characterize the species, which is being reported for the first time in Romania.

## 2. Materials and Methods

### 2.1. Study Area

This study was carried out between December 2021 and January 2023 on 72 *D. dama* (36 males and 36 females) aged between one year and 10 years from 11 hunting grounds in Arad County, Romania. The animals were hunted in accordance with the annual harvest quotas set by the Ministry of Environment, Water, and Forestry. The establishment of these quotas was carried out on the basis of hunting management criteria and followed the extraction of *D. dama* specimens by sex, specimen quality, and age categories [30]. Afterward, feces and liver samples were taken from each individual and examined in the Parasitic Diseases Clinic of the Faculty of Veterinary Medicine/University of Life Sciences “King Mihai I” from Timisoara, Romania.

Phyto-climatically, the favorable habitat for the *D. dama* found in Arad County is a typical forest steppe with small local variations depending on the climatic, soil, and vegetation factors that create zonal peculiarities. The area of the lower meadow of Mureș is characterized by favorable habitats for *D. dama*, consisting of briar and sedge forests with rich forest vegetation [36]. The Sâmpetru Gheduș hunting ground falls within the general characteristics of this habitat.

In the forest steppe and steppe areas of the Banat and Arad plains, there were habitats of Ponto-Pannonian thickets [36]. The Socodor, Râtu Pil, Lunca Holumburi, Balta, and Pâncota hunting grounds fall within these usual characteristic Danubian–Balkan forests of *Quercus cerris* and *Quercus fraineto* (Figure 1) [36].

In the county of Arad, in the spring of 2023, a herd of 2852 *D. dama* was assessed, and on the 11 hunting grounds included in this study, a herd of 2574 *D. dama* was assessed [33]. In the 11 hunting funds, wild populations are exclusively free. *C. elaphus*, *C. capreolus*, and other species of ungulates such as *S. scrofa* coexist with *D. dama* establishing various interspecific relationships between them. These habitats are also used by domestic animals (*O. aries*, *C. aegagrus hircus*, *B. taurus*) belonging to local communities, with the certainty of indirect contact by using the same areas of hay and pasture, or directly, by contact between domestic and wild animals.

### 2.2. Coprological Examination

Fecal samples were collected from the gut of each individual and stored at 4 °C until processing. From each thoroughly mixed fecal sample, a subsample, 5 g of feces, was collected and analyzed using the quantitative sedimentation technique using a sedimentation time of 30 min [37]. The samples were examined under a microscope at 40× magnification.

### 2.3. Necropsy Examination of the Liver

The liver from each *D. dama* under study was examined macroscopically for shape, size, irregular formations, fibrin deposits, translucency of the Glisson capsule, and traces of iron porphyrin. For proper examination, each liver was sectioned into slices of approximately 2 cm, and during the section, light pressure was applied to the organ. The slices were carefully examined from both sides and subsequently placed in a tray of water for several hours following the technique proposed by Zajac and Conboy [37]. Parasites were counted, measured, and stored in 96% ethanol for molecular analysis.

### 2.4. PCR Protocol

DNA was extracted from the adult fluke. Extraction was performed using the ISOLATE II Genomic DNA Kit (BIOLINE^®^ UK Ltd., London, UK). During successive stages, cellular components of a different nature were eliminated, so that at the end of the process, the DNA was precipitated, thus obtaining purified DNA without RNA contamination. The DNA was stored in a freezer at −20 °C until further analysis by PCR. The PCR reaction was performed according to the technique described by Lotfy et al. [38], with some minor modifications. Amplification of the ITS2 region was performed by classical PCR and was based on the creation of multiple copies of a ~500 bp gene sequence using the forward GA1 (5′-AGA ACA TCG ACA TCT TGA AC-3′) and reverse BD2 (5′-TAT GCT TAA ATT CAG CGG GT3′) primers.

The second reaction was performed according to the procedure described by Cucher et al. [39] using the specific primers FhCO1F: 5′-TAT GTT TTG ATT TTA CCC GGG-3′ and FhCO1R: 5′-ATG AGC AAC CAC AAA CCA TGT-3′, which amplify a 405 bp fragment of cytochrome c oxidase subunit 1 (COX-1) genes of *F. hepatica*.

Amplification was achieved after the protocol described using Master Mix MyTaqTM Red Mix (BIOLINE^®^ UK Ltd., London, UK). The final volume of the PCR amplification was 25 µL, which consisted of 12.5 µL of MyTaqTM Red Mix, 1 µL of forward and reverse primers (diluted to a concentration of 10 pmol/µL, according to the protocol described by the manufacturer), DNA extracted from the sample, and ultrapure PCR water. For the PCR reaction, 1 µL of a DNA solution of 406.7 ng/µL was used.

The amplification was performed with the thermocycler My Cycler (BioRad^®^, Berkeley, CA, USA). The program included DNA denaturation at 95 °C for 1 min; 32 cycles of denaturation at 95 °C for 30 s; hybridization at 49 °C for 30 s; extension at 72 °C for 30 s; and final extension/elongation 10 min at 72 °C, followed by incubation at 4 °C.

The amplicons were visualized by horizontal electrophoresis in a submerged electrophoresis system in 1.5% agarose gel, with the addition of the fluorescent dye RedSafe™ (iNtRON Biotechnology, Inc., Gyeonggi-do, Republic of Korea), at a voltage of 120 V and 90 mA, for 60 min. The 100 bp DNA ladder marker (BIOLINE^®^ UK Ltd., London, UK) was used. The image of the gel with the migrated DNA fragments was captured using a UV photo documentation system (UVP^®^, LLC, Upland, CA, USA). The samples were cleaned using the commercial kit ISOLATE II PCR and Gel Kit (Bioline, London, UK) according to the manufacturer’s protocol and sent to be sequenced.

To confirm the results, both PCR products were sequenced in the forward and reverse direction by the company Macrogen Europe B.V., Amsterdam, The Netherlands. A homology search was performed using the online version of the Basic Local Alignment Search Tool (BLAST) software (available at: https://blast.ncbi.nlm.nih.gov/Blast.cgi; accessed on 29 January 2023).

Phylogenetic analyses using the ITS gene sequences isolated from samples in the present study as well as those of the same genus retrieved from the GenBank databases were used. The accession number of the sequences analyzed in the present study is colored red in the figure, showing the phylogenetic tree (Appendix A). Multiple alignments of the nucleotide sequences of the haplotypes were performed using the Clustal W algorithm. A maximum likelihood (ML) analysis was performed with the program PhyML [40,41] provided on the ‘phylogeny.fr’ website (available at: http://www.phylogeny.fr/, accessed on 28 February 2024) and visualized and annotated in iTOL v6 [42].

## 3. Results

From the 72 fecal samples examined using the sedimentation method, in four samples, we highlighted the presence of fluke eggs (5.56%). The *D. dama* from which the positive fecal samples were collected came from the Sâmpetru Gheduș hunting ground in Arad County. Morphologically, the type of egg identified belongs to the fluke *F. magna* (Figure 2).

From the 72 liver samples from *D. dama*, we identified adult flukes in only four samples (5.56%) (Figure 3). The specimens from which the positive liver samples were collected were obtained from the Sâmpetru Gheduș hunting fund in Arad County.

Liver lesions varied according to the disease stage. In sample one, severe and extensive liver changes were observed throughout the organ: an irregular surface, traces of dark pigment, and nodular protrusions (hemorrhages) (Figure 4). Six pseudocysts with hemorrhagic contents and 39 adult flukes were identified. Each of these formations harbored five, six, five, seven, four, and five flukes, respectively, with dimensions between 4 and 8.2 cm. Seven flukes were quantified along the migratory tunnels. The second sample presented with lesions of fibrinous interstitial hepatitis alternating with hemorrhages (Figure 5). Four pseudocysts with hemorrhagic contents and 24 adult flukes were identified. Each pseudocyst presented five, seven, five, and three flukes with dimensions between 3.8 and 8 cm. Four flukes were present along the migratory tunnels. In the third sample, cases of mild liver damage were observed, as well as one pseudocyst with hemorrhagic content, and four adult flukes with sizes between 4.1 and 8.1 cm were identified. No fluke was identified on the migratory tunnels. In the fourth sample, localized liver lesions were observed. Two pseudocysts were identified that contained a hemorrhagic fluid and seven and four adult flukes, respectively, with dimensions between 4 and 8 cm. No fluke was identified on the migratory tunnels. The four positive samples came from *D. dama* specimens collected on 10 February 2022, 13 October 2022, 19 October 2022, and 25 October 2022.

PCR amplification revealed clear bands at ~500 bp for GA1/BD2 primer set amplifications and at a 405 bp for FhCO1F/FhCO1R primers. The samples were cleaned and sent to sequencing (Appendix A).

Sequencing showed that the trematodes identified in the *D. dama* were similar to the sequence of *F. magna* (GenBank no. EF534992.1, DQ683545.1, KU232369.1). The sequence obtained from the molecular analysis has been deposited in GenBank^®^ under accession number OQ689976.1 (Appendix A). The phylogenetic analysis in Supplementary File **1** reveals the homology with other *F. magna* isolates like KU232369, EF051080, EF612487, DQ683545, and EF534992, identified in wild or domestic ruminants from different parts of the world. As an outgroup, in the phylogenetic analysis, the isolates AB553816 and AB553823 of *Fasciola hepatica* were used.

## 4. Discussion

The diagnosis of fascioloidosis established in 4 out of 72 *D. dama* (5.56%) from the Sâmpetru Gheduș hunting reserve (Arad County) in the Western Plain of Romania is supported by the following elements: the macroscopic appearance of liver lesions, the morphological characteristics of the parasite identified in the lesions, the presence of eggs in the feces collected from the intestines of the examined cervids, and characteristics of the habitat where the samples came from, favorable for both the definitive and the intermediate host. The diagnosis of the species was undoubtedly confirmed by multiplex PCR, with the species *F. magna* being, for the first time, described molecularly in *D. dama* from Romania.

An overview of the European continent helps us to understand fluke expansion from the three initial foci (northern Italy, the south and center of Bohemia, and the Danube area) to other countries where cervids (*C. elaphus*, *D. dama*, *D. dama*) are affected, as well as domestic ruminants: Poland, Slovakia, Austria, Germany, Croatia, Hungary, and Serbia [3,7,17,43,44,45,46,47].

*Fascioloides magna* is considered an invasive species in Europe and is defined as exotic, even alien. The introduction and spread of the fluke threaten both biological diversity and the health of the final hosts [45]. We notice the presence of the fluke in the countries neighboring Romania, Hungary, and Serbia, and the *F. magna* identification species for the first time, which was molecularly confirmed in *D. dama* from the Sâmpetru Gheduș hunting ground in Arad County (Romania), located on the border with Hungary.

The recent occurrence of *F. magna* in wild ungulates (*C. elaphus*, *D. dama*, *C. capreolus*, sika deer) in the area of the Upper Palatinate Forest in northeastern Bavaria provides epidemiological evidence for the spread of the parasite in Germany by migratory cervids [48].

The first identification of *F. magna* in *D. dama* from Poland was made by Karamon et al. [17]. One year later, Houszka et al. [49] drew attention to the risk that *F. magna* infection presents for wild cervids, in the context in which the authors highlighted the fluke in the liver lesions of *C. capreolus*, *C. elaphus*, and *D. dama*.

The migration of the *C. elaphus* between the two existing foci (southwest and southeast areas), the spread of the infected aquatic snails, and the larval forms of the parasite in the watercourses and rivers make *F. magna* infection a threat to cervids and domestic ruminants in Poland [50,51,52].

The study conducted by Sattmann et al. [45] revealed, in individuals of *F. magna* harvested from the north and south of the Danube (Austria), the presence of the concatenated Ha5 haplotype, as the most widespread haplotype in Europe, which Králová-Hromadová et al. [53] claim to exist in the Czech Republic, Slovakia, Hungary, Poland, and Croatia. Husch et al. [54] supported the homogeneity of the genetic diversity of *F. magna* in Austria (concatenated haplotype Ha5) compared with the Czech Republic and Slovakia, where other haplotypes are also present [18]. *Fascioloides magna* has been reported in Croatia in *C. capreolus* [3], *O. ammon musimon* [19], *S. scrofa* [24], and *C. elaphus* [55]. It is speculated that the importation of *F. magna* into Europe induced not only host animal adaptations but also fluke adaptations to new intermediate and final hosts [3,46]. 

The presence of *F. magna* has been reconfirmed in Serbia [47], eight years after the notification made by Marinkovic and Nesic [13]. The diagnosis of fascioloidosis in the *D. dama* population in Serbia is the result of the natural spread of the fluke originating from deer specimens from Hungary and Croatia along the Danube River [47]. Prevalences similar to those observed in European natural foci have been described by Hungarian authors who hypothesize that the occurrence of *F. magna* in *D. dama* and *C. elaphus* originating from four neighboring regions of Hungary could be a partly natural and partly human-influenced process [56].

The examination of the four categories of hosts (native, definitive, aberrant, dead-end) led to the identification of two new haplotypes in Italy, and the Danube floodplains draw attention to the introduction of new sources of *F. magna* infection in Europe originating from the northeastern USA [57].

Špakulová et al. [28], Králová-Hromadová et al. [53], and Malcicka [16] anticipated that the giant liver fluke spreading along the Danube River is expected to appear soon in new areas where there are no barriers to the movements of its definitive hosts. Currently, the fluke is in six of the countries flooded by the Danube: Germany, Austria, Slovakia, Hungary, Croatia, and Serbia. As Hungary and Serbia are neighbors to Romania, the results of this study complete the map of fluke distribution in Eastern Europe.

The pathological changes, clinical signs, and outcome of *F. magna* infection are closely related to the type of host (definitive, aberrant, dead-end) and their tolerance to infection. The characteristic morphology of the parasite and its liver lesions support the diagnosis of this important cervid parasitosis [3,58]. In the present study, the liver lesions identified in 4 out of 72 *D. dama* varied according to the stage of the disease, from destruction and severe liver damage extended over the entire organ, irregular liver surfaces, and traces of dark pigment and hemorrhages to fibrinous interstitial hepatitis lesions that alternate with hemorrhages. The results of the present study join those reported by Karamon et al. [17] and Houszka et al. [49] in *D. dama*: fibrin traces and black pigment on the liver surface; pseudocysts filled with a brown mucous liquid and adult flukes; and serous ecchymoses present in the heart, reticulum, omasum, and abomasum, as well as fibrin on the serosa of the peritoneum and lung lesions as a result of parasite migration. 

Cervids originating from central Europe showed severe hepatic lesions in response to *F. magna* infection, as well as liver hypertrophy, and large areas of the liver surface on the diaphragm side were covered with fibrin [59]. The results of the present study revealed the presence of one to six pseudocysts that contained a hemorrhagic liquid and harbored 4–39 adult flukes with a length of 3.8–8.2 cm. In the migratory tunnels, four to seven flukes were counted. Comparatively, the morphological studies reported on red deer, *D. dama*, and *C. capreolus* evoke aspects similar to the results of the research by Novobilský et al. [43], who quantified 31 flukes in the liver of *C. elaphus* and 28 flukes in the liver of *D. dama*. One to five pseudocysts filled with one to six flukes, 5.5–8.5 cm long, and dark brown fluid along with hundreds of fluke eggs were identified in the liver of cervids examined by Filip-Hutsch et al. [59]. In the migration channels, smaller flukes with a length of 4.0–4.5 cm were found. 

*Cervus elaphus* and *D. dama* are the only host species responsible for the spread of *F. magna* in Poland. Despite this, *F. magna* infection causes severe lesions in deer [49]. It is interesting that in spite of infection with remarkable numbers of flukes, *C. elaphus* and *D. dama* usually show no clinical signs compared with *C. capreolus*, the aberrant hosts, which react with severe symptoms and even death at a low level of *F. magna* infestation, according to Houszka et al. [49].

The presence of adult flukes in the liver pseudocysts and eggs in the feces demonstrate that *C. nippon* is no longer a dead-end host for *F. magna*. It becomes a suitable definitive host of the fluke of epidemiological significance because of its involvement in the transmission of *F. magna* [58].

Fecal samples from a deer farm in northern Germany on the border with the Czech Republic were examined monthly to check for the presence of *F. magna* eggs. The prevalence was recorded as 4.9% in *C. elaphus* and 10.2% in *D. dama* [60]. In a study conducted by Hirtová et al. [61] at a deer farm in Central Bohemia, the presence of *F. magna* eggs showed values between 6% and 16%. The study performed in Poland highlighted *F. magna* eggs in the feces of *C. elaphus* and *D. dama* but did not identify eggs in the feces collected from *C. capreolus* [49].

A recent study carried out in cervids (*C. capreolus*, *C. elaphus*, *D. dama*) that live in one of the most extensive forests in the southwest of Poland (central Europe), appreciated by Polish hunters but also by those from neighboring countries for the hunting richness, revealed the presence of *F. magna* eggs in all cervid species examined, both in definitive hosts (*C. elaphus*, *D. dama*) and in aberrant ones (*C. capreolus*) [59]. Once the infection cycle is established in a habitat, fascioloidosis can hardly be controlled, even under farm conditions [60]. Considering that *D. dama* represents a notorious final host of the fluke, the highlighting of *F. magna* eggs in this study was anticipated.

Accordingly, the environment represents a decisive element in the evolution and monitoring of the giant fluke, *F. magna*. Features of the microclimate (soil type, precipitation regime, relief structure, population of intermediate hosts, etc.) associated with the size of the host population are risk factors for parasitic processes [62]. The area of the lower meadow of Mures, where the four specimens of *D. dama* parasitized with *F. magna* were diagnosed, offers a favorable habitat for both the definitive host and the intermediate one (the *G. truncatula* snail). The diagnosis of the four *D. dama* in the months of October and February, respectively, justifies the fact that this habitat allows for the development of the free stages of the fluke, and that infection is possible from spring to autumn. The environmental dynamics, typical for the floodplain forests of Mures, influence the fluctuations in the snail population and represent a testimony of the adaptability of *D. dama* to climatic and anthropogenic changes. The identification of *F. magna* eggs in the fecal samples collected from the intestine of infested *D. dama* confirms the maturity of the parasites that affected the liver and represents an epidemiological witness of the disease in this geographical area to other cervids and/or domestic ruminants.

## 5. Conclusions

If we take into consideration that *F. magna* has a high potential to infest not only cervids, but also *O. aries*, *B. taurus*, and even wild boar, monitoring the spread of the parasite and the infection risk area, as it is the habitat used by the infested *D. dama* from the Sâmpetru Gheduș hunting grounds, Arad County, Romania, is imperative. Pastures, hayfields, agricultural lands, and lands without forest vegetation are used both by species of hunting fauna (*D. dama*, *C. elaphus*, *C. capreolus,* and *S. scrofa*) and by domestic animals (*O. aries*, *C. aegagrus hircus*, *B. taurus*) belonging to local communities. These epidemiological aspects that certify the possible risk of infection of other cervids as well as domestic animals help us to imagine plausible future epidemiological scenarios for the further expansion of the liver fluke in Romania.

To the best of our knowledge, this study reports the first identification and molecular characterization of the giant liver fluke, *F. magna,* in *D. dama* in Romania. The molecular identification of the *F. magna* species in *D. dama* from the Western Plain of Romania justifies, once more, the importance of the surveillance process for the presence of *F. magna* in Europe, as well as starting cooperative actions with neighboring countries where the presence of the fluke is already confirmed. This strategy could reduce the potential negative effects of liver fluke and other invasive parasites on the biodiversity, health, and economy of host populations in Europe.

## Figures and Tables

**Figure 1 microorganisms-12-00527-f001:**
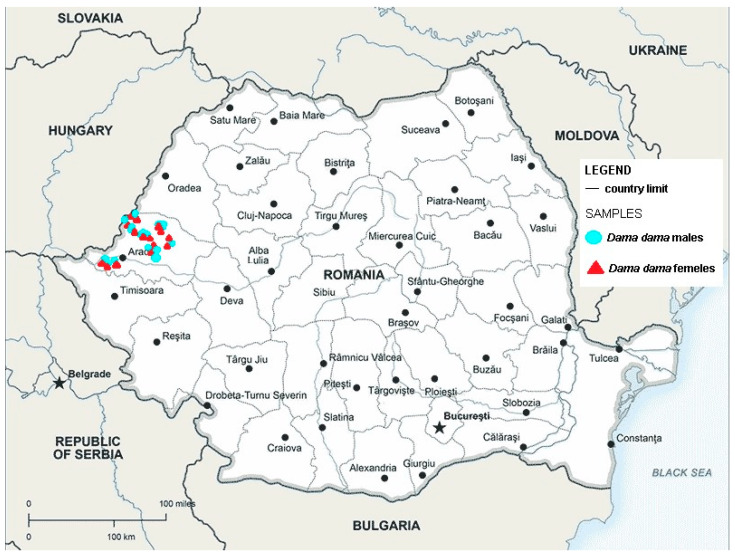
Map showing the hunting grounds of Arad county where *D. dama* were collected; red triangle shows the sites where female positive animals were found and blue circles shows the sites where male positive animals were found.

**Figure 2 microorganisms-12-00527-f002:**
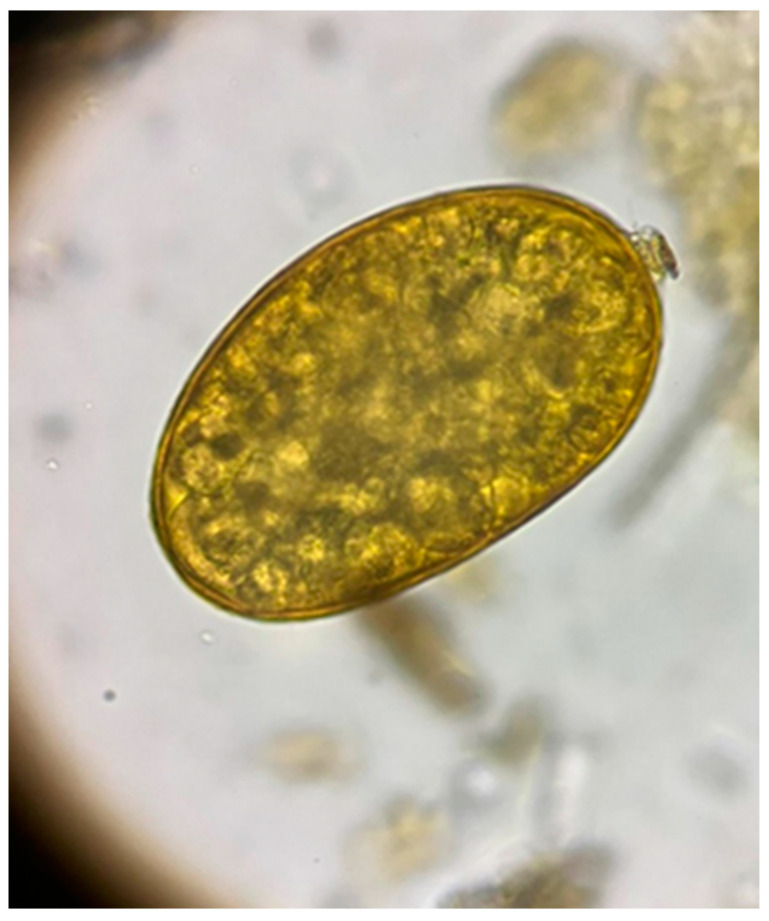
Egg of Fascioloides magna.

**Figure 3 microorganisms-12-00527-f003:**
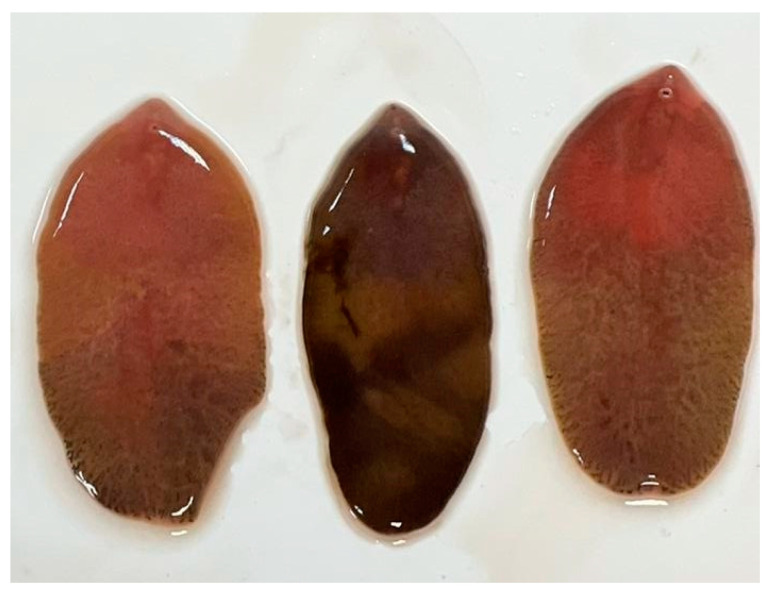
Adults of Fascioloides magna.

**Figure 4 microorganisms-12-00527-f004:**
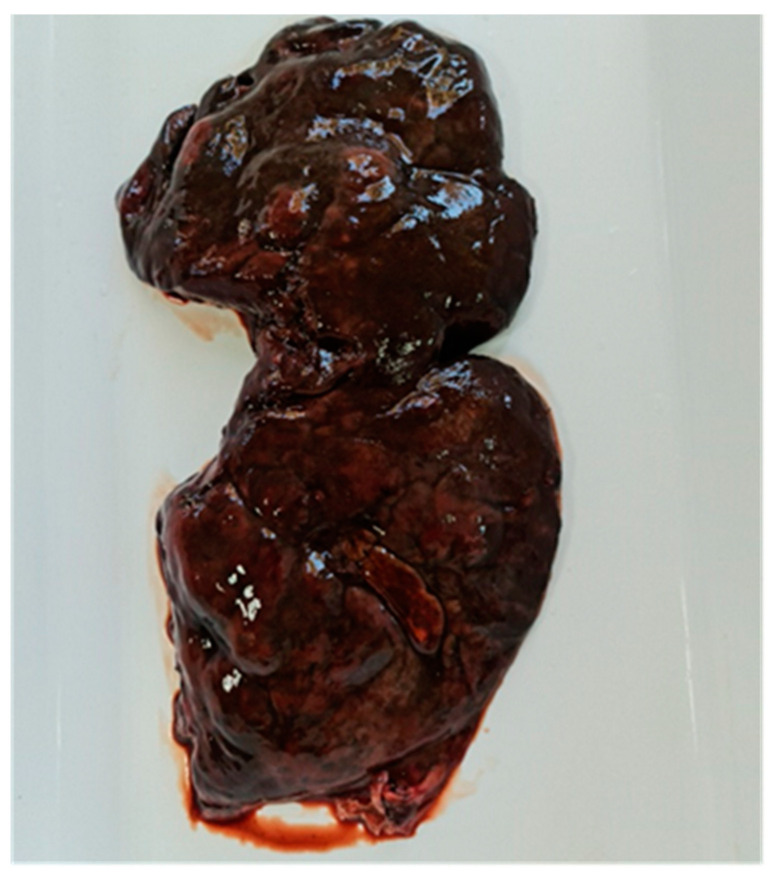
Acute infection of fallow deer liver. Irregular surface, dark pigment marks, and nodular protrusions (hemorrhages).

**Figure 5 microorganisms-12-00527-f005:**
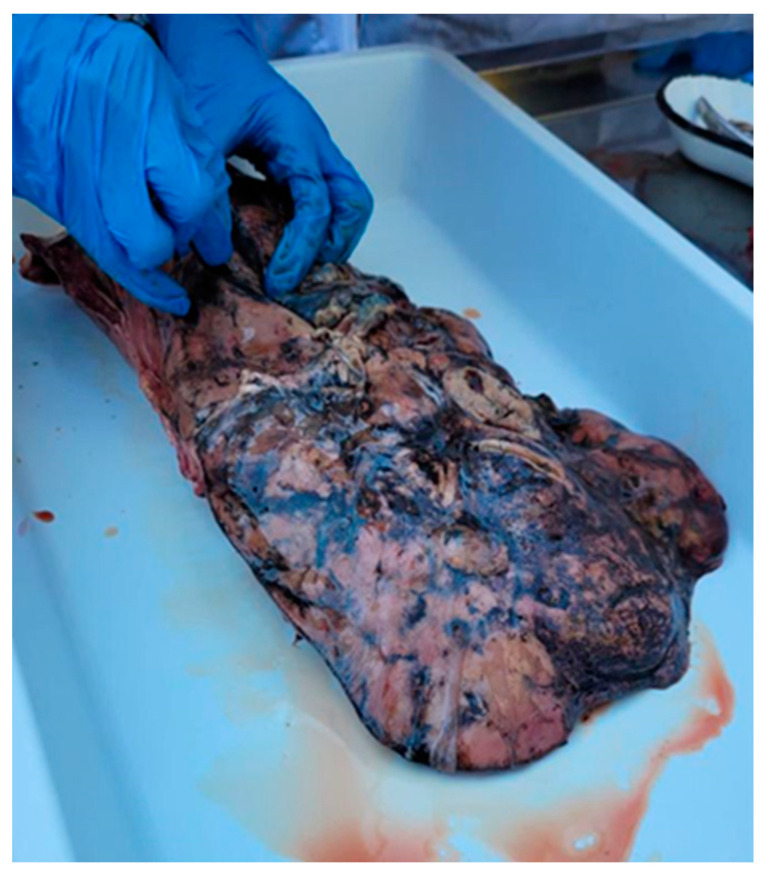
Fibrinous interstitial hepatitis lesions altering with hemorrhages.

## Data Availability

Data are contained within the article.

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
