# Peer review of "Identification and Molecular Characterization of Giant Liver Fluke (Fascioloides magna) Infection in European Fallow Deer (Dama dama) in Romania—First Report"

_microorganisms, 2024, doi:10.3390/microorganisms12030527_

Round 1

Reviewer 1 Report

Comments and Suggestions for Authors

The article deals with the first finding of the trematode Fascioloides magna in the European fallow deer Dama dama from hunting grounds in Arad County (Romania). Continuance of the molecular study epic by Romanian authors of the parasites from the European fallow deer. It should be noted that this manuscript is designed and structured much better than previous work on Gongylonema pulchrum.

The merits of the article include the fact that the morphological identification of F. magna was confirmed by molecular genetic methods. Еhe authors conducted a good literature review on the theme. The article fits into scope of Microorganisms and can be published.

However, there are a few comments to the manuscript.

1. The authors presented in their article the first finding of a parasite species in a host in a certain area. In such cases, in addition to molecular diagnostics, a morphological description of the parasite and a drawing are usually provided. Or at least a high-quality photo of a stained parasite. Figure 3 doesn't count, it's just terrible. Of the 39 trematode specimens found, is this really the best?

2. Also, I would like to see in the article a traditional description of the parasite with morphological feathures of Fascioloides magna, and not just the length of trematodes.

3. Chapter “PCR Protocol” repeats the previous article almost word for word, although is written more accurately and scientifically. But there are still comments:

lines 156-162 - Remove the entire paragraph! Accordingly, remove the photo of the phoresis too. This is not needed in this article.

lines 145,146 - The statement is unscientific. Should be rephrased.

lines 150,151 –  “… and ultrapure water.” -  What does this even mean?

line 152 – “The amplification program was performed with …”  – What does this even mean?

line 155 – “…followed by incubation at 4°C.” – What's this?  PCR-programm without final elongation?

4. The first mention of the species in the text. It is necessary to provide the author of the description and the year of the description. This is still the main character of the article. It also wouldn’t hurt to give the names of the order and family of the parasite at the first mention.

5. In captions to figures, it is better to give Latin names in full.

6. Lines 42, 220, 244 – A sentence cannot begin with an abbreviated word. In this case, you must write the generic name in full.

7. Yet again correct name - “the European fallow deer”, at least in the title of the article.

8. lines 54,128, 136, 231, 238, 241, 242, 249, etc. Again, when referring to an article in the text, the year and authors’ initials are not indicated (for example, Hasse et al. [27]).

9. Line 163 – “To confirm the trematode validation …”?

10. it looks better (line 223): “… and the F. magna identification species for the first time …”

11. Line 191 – All numbers at the beginning of a sentence are written in words (“Two pseudocysts ..”)

And in general, numbers from 1 to 9 in articles are written in words. Please, correct the text.

12. First mention of species in the text – Latin name in full needed.

13. According with the MDPI rules in Article title all words must be capitalized (except for the fluke name).

The manuscript can be published in Microorganisms, but minor corrections are needed.

Author Response

Thank you very much for your observations, they were very welcome and made us reorganize our paper. We upload the entire paper for you to reconsider for publishing as a Word document, where all corrections are suggested. We have used the track changes application, as suggested.

Reviewer 2 Report

Comments and Suggestions for Authors

The study by Popovici et al. 2024 presented data on the molecular analysis of Fascioloides magna from fecal samples and livers of fallow deer in Romania. The authors claim that the reported data is the first in Romania which adds to the innovation of the study. However, the data presentation in the result and discussion could be improved. The authors could perform phylogenetic analyses with the obtained sequences of F. magna in order to show where they are situated among the sequences available in the GenBank. The structure of the discussion could be improved. Moreover, the authors could make a clear conclusion based on the results. Furthermore, the paper need editing for language to improve clarity.

Comments on the Quality of English Language

The language and punctuation need to be edited before being published.

Author Response

(The authors gave the same response as above.)
